# Alterations in Bacterial Metabolism Contribute to the Lifespan Extension Exerted by Guarana in *Caenorhabditis elegans*

**DOI:** 10.3390/nu14091986

**Published:** 2022-05-09

**Authors:** Inés Reigada, Karmen Kapp, Claire Maynard, David Weinkove, Marta Sofía Valero, Elisa Langa, Leena Hanski, Carlota Gómez-Rincón

**Affiliations:** 1Drug Research Program, Division of Pharmaceutical Biosciences, Faculty of Pharmacy, University of Helsinki, 00014 Helsinki, Finland; karmen.kapp@helsinki.fi (K.K.); leena.hanski@helsinki.fi (L.H.); 2Department of Biosciences, Durham University, Durham DH1 3LE, UK; ca_maynard@outlook.com (C.M.); david.weinkove@durham.ac.uk (D.W.); 3Department of Pharmacology, Physiology and Legal and Forensic Medicine, University of Zaragoza, 50009 Huesca, Spain; msvalero@unizar.es; 4Instituto Agroalimentario de Aragón-IA2 (CITA-Universidad de Zaragoza), 50013 Zaragoza, Spain; cgomez@usj.es; 5Department of Pharmacy, Faculty of Health Science, Universidad San Jorge (San Jorge University), 50830 Zaragoza, Spain; elanga@usj.es

**Keywords:** *Paullinia cupana*, nutraceuticals, microbiota, *Escherichia* *coli*, folate, aging, dietary supplement, antioxidant, inflammatory bowel disease, proteobacteria

## Abstract

Guarana (*Paullinia cupana*) is a widely consumed nutraceutical with various health benefits supported by scientific evidence. However, its indirect health impacts through the gut microbiota have not been studied. *Caenorhabditis elegans* is a useful model to study both the direct and indirect effects of nutraceuticals, as the intimate association of the worm with the metabolites produced by *Escherichia* *coli* is a prototypic simplified model of our gut microbiota. We prepared an ethanoic extract of guarana seeds and assessed its antioxidant capacity in vitro, with a 2,2-diphenyl-1-picrylhydrazyl (DPPH) assay, and in vivo, utilizing *C. elegans*. Additionally, we studied the impact of this extract on *C. elegans* lifespan, utilizing both viable and non-viable *E. coli*, and assessed the impact of guarana on *E. coli* folate production. The extract showed high antioxidant capacity, and it extended worm lifespan. However, the antioxidant and life-extending effects did not correlate in terms of the extract concentration. The extract-induced life extension was also less significant when utilizing dead *E. coli,* which may indicate that the effects of guarana on the worms work partly through modifications on *E. coli* metabolism. Following this observation, guarana was found to decrease *E. coli* folate production, revealing one possible route for its beneficial effects.

## 1. Introduction

In the last few decades, the consumption of dietary supplements and nutraceuticals has dramatically increased [1]. The popularity of many nutraceuticals is based on the health benefits granted to them by traditional use, guarana (*Paullinia cupana*, Sapindaceae) being an excellent example. The seeds of this evergreen climbing shrub have been consumed for centuries by the native Brazilian population [2], who used the seeds as a stimulant to hunt, as a potent diuretic and also to reduce fever and headaches [2]. Currently, guarana is sold as a dietary supplement in the form of tablets and capsules, but it is also often found as part of energy drinks, given the increasing preference of consumers for natural products [3].

The pharmaceutical properties of guarana have been extensively studied by the research community, and many of the obtained results provide evidence for its antioxidant [4], anti-inflammatory [5], neuroprotective [6], anti-fatigue [7], hypocholesterolemic [8], and anti-obesity effects [9,10]. However, as with many other oral supplements, the impact of guarana on bacterial metabolism and, therefore, its potential impact on the microbiota, has been less explored.

Diet and dietary supplements can directly or indirectly affect the human body. The direct effects involve those caused by the absorbance of nutrients and compounds from the diet itself, while the indirect dietary effects are those mediated by the microbial community in the digestive track [11]. The enormous impact that the microbiota composition and stability have on human health is currently well known, and the role of diet in significantly affecting the functionality of the microbiota is generally acknowledged [12]. For instance, it has been observed that the gut microbiota of the elder population and patients with inflammatory bowel diseases (IBD) are characterized by an overgrowth of proteobacteria and increased serum folate levels [13,14]. These bacteria are able to synthesize folate de novo, which has been hypothesized to increase folate-dependent toxicity, which may damage tissues, promote inflammation and accelerate aging [12]. Thus, it would be relevant to evaluate the impact of nutraceuticals and dietary supplements on bacterial metabolism, for instance, assessing their impact on proteobacteria folate production [15,16].

*Caenorhabditis elegans* has proven to be a very useful model organism to scientifically assess the properties and safety of nutraceuticals, as it manages to shorten the translational gap between the in vitro and the in vivo conditions, avoiding the ethical concerns of other animal models. Besides, their relatively short lifespan, the easy propagation of populations of synchronized individuals, and their genetics, make *C. elegans* a powerful model for the study of aging and health-promoting effects of exogenous substances [17,18].

In fact, *C. elegans* has been previously used to shed some light on the mechanism of action insights behind the antioxidant, anti-aging, and neuroprotective capacity of guarana [19,20,21]. Moreover, *C. elegans* has proven to be a supreme tool for the study of microbe–host relations, being a suitable model for the assessment of the molecular processes involved in pathogenic and commensal interactions [22]. The usefulness of this model lies in the fact that these molecular processes are often conserved in other organisms of interest, such as humans [22]. For instance, at the intestinal level, there are morphological similarities in the intestinal cellular structure between *C. elegans* and humans, enabling comparison of extraction functionalities and absorption of nutrients [22]. Besides, bacteria can also colonize the intestine of *C. elegans* and this association may benefit the host, as it happens with the human microbiota [23]. In laboratory conditions, *C. elegans* are often exposed exclusively to the Gram-negative bacterium, *Escherichia coli*. During development and in young adult worms, the interaction is primarily nutritional, *E.coli* acting as the main nutrient of the worm [16]. However, upon aging, *E. coli* colonize the intestine of the worm, shifting the host–microbe relationship. While *E. coli* does not constitute standard microbiota, the intimate association of the worm with the metabolites produced by this bacterium throughout its lifespan provides a relevant model to study the impact on the variations of the bacterial metabolism on the health of the host [16]. For instance, it has been previously observed that the inhibition of *E. coli* folate production slows *C. elegans* aging [24,25]. This could make *C. elegans* an interesting model to identify potentially beneficial nutraceuticals in pathological processes, characterized by microbiota dysbiosis and increased bacterial folate production. 

Therefore, in this study, we utilized the in vivo model, *C. elegans,* to characterize the properties of guarana, not only by assessing its direct effects on the worm, but also through its impact on *E. coli* metabolism. To do so, we chemically characterized an ethanolic extract of guarana seeds and we assessed its antioxidant capacity in vitro and in *C. elegans.* Additionally, we assessed the impact of this extract on the worm lifespan and to assess if the extract effects were given directly to the worm, or throughout its impact on the bacterial metabolism, and we utilized both viable and non-viable *E. coli*. Finally, we assessed the impact of guarana on *E. coli* folate production.

## 2. Materials and Methods

### 2.1. Plant Material and Extraction

Dry guarana seeds were provided by Plantarom^®^ (Montreal, QC, Canada) and stored at room temperature protected from humidity and direct light until use. The seeds were crushed to homogeneous particle size. The crushed guarana seeds (50 g in total) were introduced to a Soxhlet apparatus and the extraction was performed with ethanol (96%) for 4 h [26]. The extract was evaporated to dryness with a rotary flash evaporator (Büchi^®^ Rotavapor^®^ R-210, Büchi laboratories, Flawil, Switzerland) and the extract was preserved at −20 °C until further use.

### 2.2. UPLC-PDA-MS Analyses

Guarana seed extract was dissolved in 50% aqueous ethanol for the stock solution of 5.0 mg/mL. The extract was vortexed, sonificated at RT for 15 min and filtered through 0.45 µm polypropylene syringe filter (VWR, Radnor, PA, USA). For the UPLC-MS analyses, a working solution with a concentration of 100 µg/mL was prepared. Identification of compounds was achieved using analytical reversed-phase UPLC-MS system.

The chromatographic system consisted of Acquity UPLC (Waters, Milford, MA, USA) equipped with a photo-diode array (PDA) detector (Waters, Milford, MA, USA) coupled to a mass single-quadrupole detector (QDa, Waters) with an electrospray ionization (ESI) interface. Empower software v. 3.0 was used for instrument control and data acquisition. Samples were separated on an Ascentis^®^ Express HPLC column (50 mm × 2.1 mm, 2.0 μm, Supelco, Beliefonte, PA, USA). Column temperature and sample injection volume were, respectively, set to be 30 °C and 2.0 μL.

With the PDA detector, the mobile phase comprised a mixture of 15 mM KH_2_PO_4_ pH 2.0 (solvent A, Sigma-Aldrich, St. Louis, MO, USA) and acetonitrile (solvent B, ChromasolvTM, Honeywell Research Chemicals, Morris Plains, NJ, USA). A 10-min gradient program was used, beginning with 10% B, increasing to 80% at 8 min and a decrease to 10% B, holding for 2 min. The flow rate of the mobile phase was 0.5 mL/min. Compounds were monitored with Max plot chromatogram.

For the UPLC-MS runs, the mobile phase comprised a mixture of acetonitrile (solvent A, ChromasolvTM, Honeywell Research Chemicals, Morris Plains, NJ, USA) and 0.1% *v/v* formic acid (solvent B, Merck, Darmstadt, Germany) in water. An 8-min gradient program was used, beginning with 5% B, increasing to 70% at 7 min and a decrease to 5% B, holding for 1 min. The flow rate of the mobile phase was 0.5 mL/min. The QDa conditions were set as follows: a cone voltage of 45 V, a capillary voltage of 0.8 kV, and a source temperature of 600 °C. Runs were performed with ESI interface working in negative ionization and positive ionization, acquired in the range of 150–800 Da.

### 2.3. Strains and Culture Conditions

The *C. elegans* strains, N2 (wild type) and SS104 *glp-4*(*bn2*) used in this study were obtained from Caenorhabditis Genetics Center (CGC), Minneapolis, MN, USA. The SS104 *glp-4*(*bn2*) strain was used as the inserted mutation causes the worms not to produce offspring at a temperature of 25 °C, which facilitates the counting and handling of lifespan assays.

Both strains were maintained in Nematode Growth Media (NGM) agar plates seeded with a lawn of *Escherichia coli* OP50 (CGC, Minneapolis, MN, USA). Miller’s lysogeny broth and agar (LB; Fisher BioReagents, Pittsburg, PA, USA) were used for maintenance and growth of *E. coli*. To generate the bacterial lawn on NGM, 80 µL of a pre-culture of *E. coli* (10 μL of a glycerol stock in 5 mL of LB incubated for 18 h at 37 °C, 120 rpm) was seeded on the agar plates and incubated overnight at 37 °C. NGM medium was prepared as described [27], using 0.003 g/mL NaCl; 0.0025 g/mL peptone; 0.017 g/mL high-purity agar; 0.1% Cholesterol; 0.1% MgSO_4_ (1M); 0.1% CaCl_2_ (1M), and 2.5% of Potassium Buffer (1M) (all purchased from Sigma-Aldrich, St. Louis, MO, USA). The wild-type strain was routinely maintained at 20 °C while SS104 was maintained at 15 °C.

### 2.4. Antioxidant Activity In Vitro: DPPH Method 

The free radical activity of the extract was evaluated by the 2, 2-diphenyl-1-picrylhydrazyl (DPPH; Sigma-Aldrich, St. Louis, MO, USA) method following the protocol of López, et al. [28]. The extract was serially diluted in ethanol (concentrations ranging from 1 to 500 µg/mL) and 150 µL of each dilution was mixed with 150 μL of a DPPH ethanolic solution (0.04 mg/mL). Epicatechin and the positive control, ascorbic acid (Sigma-Aldrich, St. Louis, MO, USA), were used at concentrations ranging from 1 to 100 µg/mL. After 30 min of incubation (room temperature, in the darkness) the absorbance was measured at 517 nm [29]. The control samples consisted of all the reagents excluding the plant extract. The background interferences from solvents were deducted from the activity values of the corresponding extracts prior to calculating the percentage of Radical Scavenging Capacity (%RSC) as follows: % RSC = ((Abs_control_ − Abs_sample_)/Abs_control_) × 100, where Abs_control_ is the absorbance of the control and Abs_sample_ is the absorbance of the sample.

The 50% effective concentration (EC_50_), in this case, the concentration effective in producing 50% of the maximal RSC, was calculated for the guarana extract and the positive control, ascorbic acid.

### 2.5. Stress-Resistance Assays

For the stress-resistance assays, the protocol of Reigada, et al. [30] was followed. Briefly, eggs were prepared by bleaching adults and incubated in M9 buffer (6 g/L Na_2_HPO_4_; 3/L g KH_2_PO_4_; 5 g/L NaCl; 0.25 g/L MgSO_4_ × 7H_2_O in MQ water; all reagents from Sigma-Aldrich, St. Louis, MO, USA) until the hatching of the eggs (L1 stage). L1 larvae were incubated in NGM containing *E. coli* OP50 and different concentrations of guarana (100, 250 and 500 µg/mL) until L4 stage. Non-supplemented NGM plates were used as controls. After the incubation period the adults were transferred to fresh NGM plates containing 150 µM juglone (5-hydroxy-1,4-naphthalenedione; Sigma-Aldrich, St. Louis, MO, USA). Resistance to oxidative stress was assessed by counting the alive worms after 24 h incubation at 20 °C [30]. This was done by a touch-provoked movement by which worms that reacted to the mechanical stimulus were scored as alive whereas non-responding worms were considered to be dead. The survival rate (SR) was calculated as a percentage: SR (%) = (N° of worms alive × 100)/Total worms in the well.

### 2.6. Measurement of Catalase (CAT) and Superoxide Dismutase (SOD) Activity 

L1 larvae were incubated in *E. coli* OP50-seeded NGM (control) or NGM containing guarana (100 µg/mL) until L4 stage. This guarana concentration was selected as it was the lowest concentration increasing *C. elegans* survival in the stress-resistance assay. After incubation, for the protein extraction, worms were suspended in cold buffer (150 mM NaCl, 50 mM Tris-HCl pH 8, and 1% TWEEN 20 at 4 °C; all reagents from Sigma-Aldrich, St. Louis, MO, USA) and sonicated in a water bath sonicator for 5 min at 35 kHz (JP Selecta 3000683, J.P. Selecta, Barcelona, Spain). The concentration of total protein in homogenized worms was measured using BCA kit (Thermo Fisher Scientific, Waltham, MA, USA) and adjusted to the specifications of the CAT and SOD activity quantification kits (Cayman Chemicals, Ann Arbor, MI, USA), which were used for the CAT and SOD activity determination.

### 2.7. Quantification of Total Glutathione in C. elegans

For this assay, the protocol of Jenkins, et al. was followed [31] with some modifications. The worms were incubated in the same conditions as descried in Section 2.5, after which 50 adults per condition were collected in 200 µL of S-basal (5.85 g/L NaCl, 1 g/L K_2_HPO_4_, 6 g/L KH_2_PO_4_, 5 mg/L cholesterol; all reagents from Sigma-Aldrich, St. Louis, MO, USA) and transferred to 1.5 mL Eppendorf tubes. Two wash cycles were conducted with S- basal after which the worms were spun down and the total volume reduced to 20 µL. Later, a 50 µL aliquot of extraction buffer was added and the tubes were frozen in liquid nitrogen. The extraction buffer consisted of: 6 mg/mL 5-sulfosalicylic acid dehydrate, 0.1% *v/v* Triton X-100 and Complete, and Ethylenediaminetetraacetic acid (EDTA)-free Proteinase inhibitor cocktail in KPE buffer (0.1 M potassium phosphate buffer and 5 mM EDTA at pH 7.5). After freezing, the samples were homogenized in a water bath sonicator (Ultrasonic Cleaner 3800 water bath, Branson Ultrasonics, Danbury, CT, USA) at 35 kHz and cooled to 4 °C, using 10 cycles of 10 s. The samples were later centrifuged at 14,000× *g* at 4 °C and the supernatants collected.

For the measurement of the total glutathione (GSH) the protocol of [32] was followed. As such, 83.3 units/mL of glutathione reductase (GR; Sigma-Aldrich, St. Louis, MO, USA) was added to the sample with 666 μg/mL of 5.5-dithio-bis (2-nitrobenzoic acid) (DTNB; Sigma-Aldrich, St. Louis, MO, USA). After 30 s, 666 μg/mL of β- nicotinamide adenine dinucleotide phosphate (β-NADPH, Sigma-Aldrich, St. Louis, MO, USA) was added and the formation of 5′-thio-2-nitrobenzoic acid (TNB) chromophore product was recorded using a Multiskan Sky microplate spectrophotometer (Thermo Fisher Scientific, Waltham, MA, USA) at 412 nm.

### 2.8. Lifespan Analysis

Gravid adults of the *C. elegans* strain SS104 were used to lay eggs onto fresh NGM OP50 plates. Eggs rose at 15 °C until L3/L4 stage, due to temperature sensitivity of mutant phenotypes. Once this stage was reached, they were transferred to 25 °C incubation. After 24 h at 25 °C, *C. elegans* were transferred to five replicate plates for each condition. Effects on longevity of three different concentrations of guarana were measured in this analysis: 100, 250 and 500 µg/mL and NGM plates were used as a control. Animals were transferred to fresh plates after 7 and 14 days and scored for survival every 2 days [27]. Scoring method was the same used as for the oxidative stressed induced by juglone method. For the experiments with dead *E. coli*, the exact same protocol was followed, but before the transfer of the worms to the agar plates, the *E. coli* lawn was exposed to a UV light of 380 nm for 30 min.

### 2.9. Impact of Guarana on E. coli Viability

*E. coli* lawns were grown as described in Section 2.3 on NGM plates (control) and NGM plates containing 100, 250, and 500 µg/mL of guarana. After the incubation they were scraped and re-suspended in 1 mL of phosphate-buffered saline (PBS; 140 mM NaCl, pH 7.4). In order to assess the possible impact of guarana on the colony-forming units (CFU) of *E. coli*, serial dilutions of the suspensions (from 10^−1^ to 10^−7^) were then made in PBS and plated onto LB agar. The agar plates were incubated for 18 h and the colonies counted. Additionally, in order to measure the viability of the bacteria, 200 µL of each suspension was transferred into a 96-well plate (Thermo Scientific, Waltham, MA, USA), after which resazurin was added at concentration of 20 μM and incubated for 20 min at RT, 200 rpm. Then, the fluorescence was recorded (λ_excitation_ = 560 nm and λ_emission_ = 590 nm) using a Varioskan LUX multimode microplate reader (Thermo Fisher Scientific, Waltham, MA, USA) [33].

### 2.10. E. coli Folate Extraction 

An overnight defined media culture of *E. coli* in LB (Fisher BioReagents, Pittsburg, PA, USA) was seeded onto guarana agar plates (500 µg/mL) and NGM plates, and incubated at 25 °C for 4 days. For these experiments the peptone of the NGM was substituted by purified amino acids as described in [24]. Bacterial lawns were scraped from plates into Eppendorfs using M9 buffer and kept on ice. The bacterial concentration was assessed by optical density (OD) at 600 nm. Samples were concentrated in chilled microcentrifuge and pellets were snap frozen in liquid nitrogen. Pellets were thawed and resuspended in a volume of ice-cold 90% methanol and 10% folate extraction buffer (FEB: 50 mM HEPES, 50 mM CHES, 0.5% *w/v* ascorbic acid, 0.2 M Dithiothreitol, pH 7.85 with NaOH) in proportion to bacterial content (37.5 × OD600 × original solution volume). FEB was spiked with 10 nM methotrexate-Glu_6_ as an internal standard. Samples were vortexed vigorously and left on ice for 15 min before centrifugation in a cooled microcentrifuge (Thermo Scientific, Waltham, MA, USA) for 15 min at full speed. Supernatants were used for analysis.

### 2.11. Folate Liquid Chromatography -Tandem Mass Spectrometry (LC-MS/MS)Analysis 

The detection of the folates was performed following the protocol of Maynard, et al. [34]. Briefly, the detection was done by multiple reaction monitoring (MRM) analysis using an SCIEX QTRAP 6500 instrument (Sciex, Framingham, MA, USA). MRM conditions for 5-Me-H_4_PteGlu_3_ and 5/10-CHO-H_4_PteGlu_3_ were optimized by infusion of standards into the instrument. The optimized conditions for –Glu_3_ folates were applied to other higher folates using MRM transitions described by [35]. Further confirmation of folate identity was performed by performing enhanced product ion scans of folates of interest and comparing the fragment spectra with known standards.

The QTRAP 6500 was operated in ESI+ mode and was interfaced with a Shimadzu Nexera UHPLC system (Thermo Scientific, Waltham, MA, USA). Samples were separated using a Thermo PA2 C18 column (2.2 µm, 2.1 mm × 100 mm; Thermo Scientific, Waltham, MA, USA) with a gradient of 0.1% formic acid in water (mobile phase A) and acetonitrile (mobile phase B). Samples were maintained at 4 °C and 2 µL aliquots were injected. The column was maintained at 40 °C with a flow rate of 200 µL/min, starting at 2% B, held for 2 min, with a linear gradient to 100% B at 7 min, held for 1 min, before a 7-min re-equilibration step at 2% B that was necessary for consistent retention times. The column eluate flow to the MS was controlled via the QTRAP switching valve, allowing analysis between 4 and 8 min to minimize instrument contamination. Folates were quantified with reference to external standards of 5-Me-H_4_PteGlu_3_ and 5/10-CHO-H_4_PteGlu_3_, purchased from Schircks (Zürich, Switzerland). The matrix effects were assessed by spiking of standards into extracted samples.

The percentage of folate production was calculated using the analyte peak area of the untreated control.

### 2.12. Statistical Analysis 

The statistical tests were performed using GraphPad Prism v. 8.00. EC_50_ values were estimated by a non-linear regression. The statistical differences were always assessed against the untreated control with a Student’s *t*-test and *p* values under 0.05 were considered significant. For the lifespan data, statistical significance was determined using the Log-Rank and Wilcoxon tests [34].

## 3. Results

### 3.1. Chemical Characterization of the Guarana Extract

Based on the UPLC-PDA-MS analyses, five major compounds were tentatively identified, as shown in Table 1 and Figure 1. The first peak with protonated molecular ion [M + H]^+^ *m/z* 195.18 was identified as a xanthine alkaloid caffeine, as described by [36]. Other main compounds were identified as deprotonated ions [M − H]^−^ belonging to flavan-3-ols or its oligomers: catechin (*m/z* 289.02), epicatechin (*m/z* 289.02) and A-type procyanidin dimers (*m/z* 575.06 and *m/z* 574.96), according to [37,38].

### 3.2. Antioxidant Activity In Vitro and In Vivo

The guarana extract showed itself to be a potent antioxidant, both in vitro (Table 2) and in vivo (Figure 2). With the DPPH assay, the radical scavenging capacity shown by guarana was comparable to the one obtained with the positive control utilized, the ascorbic acid (Table 2). The high scavenging of free radicals was likely achieved by the presence of polyphenols, such as epicatechin, for which EC_50_ was also within the same order of magnitude as the positive control (Table 2). The radical scavenging capacity of the extract was also patent in vivo, where the antioxidant enzymes superoxide dismutases (SOD) and catalase (CAT) were less active when *C. elegans* was exposed to the guarana extract (Appendix A). In addition, guarana managed to significantly increase the resistance of *C. elegans* to the oxidative stress induced by juglone (Figure 2). After the exposure to a lethal dose of this pro-oxidant (150 µM), which reduced the survival of the control practically to zero, it was observed how all the tested concentrations of guarana managed to increase the survival rate (SR) when compared to the untreated control (*p* < 0.001 for all the tested concentrations). In addition, we tested if this increase in survival was due to an impact of guarana on the glutathione (GSH) system of the worm. However, guarana did not show an increase in the total GSH at any tested concentration (Appendix A).

### 3.3. Guarana Extract Increases C. elegans Lifespan

When the worms were maintained on live *E.*
*coli*, the presence of guarana on NGM resulted in a significant extension in *C. elegans* lifespan, at all the tested concentrations, 100, 250 and 500 µg/mL (Figure 3a; *p* < 0.001 in all the cases when compared to the untreated control). These effects seemed to be concentration dependent, as the life extension achieved with the highest concentration (500 µg/mL) was significantly higher than the one achieved with the lowest concentration used (100 µg/mL) (*p* = 0.003 when comparing these two concentrations with a log-rank test). We hypothesized that these effects may not only be given through the impact of guarana on the worms, but through the induced changes on the metabolites production of *E. coli*, which, as previously reported, might have a great impact on *C. elegans* survival. Therefore, to test the possibility of guarana affecting *C. elegans* lifespan by altering *E. coli* metabolism, we assessed its effects utilizing UV-killed *E. coli* (Figure 3b). In concordance with previous results, feeding the worms with non-viable bacteria increased the worm lifespan when compared to those fed with live bacteria (*p* < 0.001 when comparing the untreated controls, Appendix A) [25]. In these settings, the highest concentrations tested, 500 µg/mL, maintained the effect of prolonging *C. elegans* lifespan (*p* < 0.001 when compared to the untreated control). The concentration of 250 µg/mL still managed to extend the lifespan, but this extension was not as significant as the one achieved with live *E. coli* (*p* = 0.0011, when compared to the untreated control) and the concentration of 100 µg/mL did not show any differences with the untreated control. Overall, it seemed that the impact of guarana on extending lifespan was not as intense when the bacteria were not metabolically active. In Table 3, it can be clearly observed how the effects of guarana on life extension are not proportional when comparing the two different scenarios (alive and dead *E. coli*). For instance, at the highest concentration (500 µg/mL), the average lifespan with live *E. coli* was four and a half days longer than the control, while with dead *E. coli,* this extension was only two days.

In order to discard that the life extension observed with live *E. coli* (Figure 3a) was given due to a potential antimicrobial effect of guarana, we measured the viability of the bacteria and its growth on NGM with different concentrations of guarana. Via resazurin staining, we did not observe any reduction in the bacteria viability and the number of CFUs was not reduced at any of the tested concentrations (Appendix A).

### 3.4. Guarana Extract Reduces the Folate Production of E. coli

To analyze the impact of guarana on *E. coli* folate synthesis, we used LC-MS/MS to detect levels of individual *E. coli* tetrahydrofolates (THFs). It has been previously observed that 5-methyl THF-glu_3_ and 5/10-formyl THF-glu_3_ are among the most abundant folate species detected in *E. coli* grown in NGM [27,34]. Because of this, we decided to assess the impact of guarana on the production of these two THFs species. In Figure 4, it can be seen how the presence of guarana in NGM produced a significant reduction in *E. coli* production of 5-methyl-tetrahydrofolate and 10-formyl-tetrahydrofolate (*p* < 0.001 in both cases).

## 4. Discussion

The chemical profile of the guarana extract used in this study agreed with the one previously described in the literature. As expected, caffeine was present in the extract, as among plant species containing this alkaloid, guarana has the highest content [39]. On the other hand, the phenolic compounds found, catechin, epicatechin, and A-type procyanidin dimer, are also often detected in guarana, despite its content possibly varying, depending on the geographical location of the plant [37,40].

In agreement with previously reported results, guarana showed a high antioxidant activity in vitro [41,42,43,44,45,46]. It is not surprising that the radical scavenging capacity of the extract was so high, given the known hydrogen and electron-donating ability of polyphenols [47]. For instance, the presence of epicatechin, which showed an EC_50_ of 11.55 ± 2.23 µg/mL, might have highly contributed to the antioxidant activity of the extract. Interestingly, the EC_50_ obtained with our extract (4.696 to 8.747 µg/mL) was almost 10-times smaller than the one obtained by the guarana aqueous extract utilized by Peixoto, et al. [48] (40 µg/mL), while the EC_50_ calculated for the positive control (ascorbic acid) was the same in both studies (~2 µg/mL). The high capacity of guarana in scavenging free radicals was also patent in vivo, given that the activity of two of the major antioxidant enzymes, CAT and SOD [49], was significantly lower when the worms were exposed to the guarana extract (Appendix A). During the treatment with guarana, there is possibly a decrease in the basal levels of free radicals, which might have resulted in a down-regulation of the cellular enzymatic antioxidant defense [50].

In line with our results, Peixoto, et al.[48] also reported an increased resistance in *C. elegans* treated with guarana to the pro-oxidant juglone (Figure 2). These authors showed that when using mutant worms lacking the *daf*-16 gene, the treatment with guarana did not increase the survival rate, which could point towards an impact of guarana on the regulation of the stress resistance machinery in the worm. DAF-16 regulates the expression of several genes involved in stress resistance and longevity in *C. elegans*, being a homologue of the mammalian FOXO transcription factor [51]. Nevertheless, *C. elegans* is able to counteract oxidative stress through other pathways, such as those within the thiol redox network [52]. For instance, SKN-1, which is an orthologue of the human Nrf2, controls the regulation of genes encoding anti-oxidant enzymes, such as glutathione-S-transferase [53]. In order to assess if guarana additionally has an impact on the glutathione (GSH) system of the worm, we measured the impact of this extract on GSH pools, before and after exposure to juglone. However, no differences were found at any of the tested concentrations of guarana, while the treatment with juglone caused a decrease in total GSH levels, as expected (Appendix A) [52]. Therefore, we concluded that the antioxidant activity of guarana is not partly given through the regulation of the GSH system.

The guarana extract included in our study significantly extended *C. elegans* lifespan, at all tested concentrations (Figure 3a), which was in agreement with previous results [19,48]. Arantes, et al. concluded that the extension in lifespan produced by guarana was mediated by antioxidant activity and HSF-1, SKN-1, and DAF-16 pathways. Besides, they also demonstrated its effects through ADOR-1, which might indicate the involvement of the pyrogenic system in longevity [19]. These authors aptly suggested that the life-expanding effects of guarana might be related to the synergistic effects of the different compounds present in it, which influence the lifespan through different pathways. For instance, caffeine, also identified in our extract, has been proven to extend *C. elegans* lifespan, independently of DAF-16 and SKN-1 regulation [54]. Our results indicated that not only the synergy between the different compounds would prolong *C. elegans* lifespan, but also the combination of the direct effects of guarana on the worms and this extract’s indirect effects through its impact on *E. coli* metabolism. As shown in Figure 3b, when *C. elegans* was fed with dead *E. coli*, the impact on lifespan was not as significant as the one achieved with metabolically active bacteria. 

In recent years, it has become obvious that the microbiota have an impact on the health status of the host, including the aging process. The differences in gut microbiota composition between elderly and younger populations have been extensively proven. However, it is yet unknown if these differences are just the consequences of the changes in host physiology and diet, or if microbes could actually accelerate the aging of the host [12]. It is difficult to establish if the different changes in the microbiota are causative rather than consequential, but one of the strongest effector candidates are microbial metabolites [55]. These could act as signaling molecules to modulate the host metabolism, impact its energy balance, and potentially promote inflammation [55]. This raises the question whether nutraceuticals, such as guarana, could impact aging and pathological processes via their effects on the phenotype and metabolism of our gut microbiota. 

*C. elegans* has proven to be an ideal model to assess the indirect impact of orally administrated drugs on aging via their interactions with the bacterial metabolism. For instance, it was shown how metformin, or sulfamethoxazole increased *C. elegans* lifespan via the reduction in *E. coli* folate production [24]. For this reason, we hypothesized that guarana might have extended the worm lifespan, not only due to its antioxidant capacity but also by reducing the folate production of *E. coli,* and, indeed, we found out that it significantly reduces the production of 5-methyl THF-glu_3_ and 5/10-formyl THF-glu_3_ (Figure 4).

The process by which the inhibition of folate synthesis prevents bacteria from accelerating ageing is still unknown. Nevertheless, it has been hypothesized by other authors that *E. coli* shortens *C. elegans* lifespan through a form of toxin-based virulence, and that in bacteria, folate has functions beyond its role in biosynthetic one-carbon metabolism [24]. However, it is known that the different folate forms do not cause any effects on *C. elegans* lifespan per se [24].

In humans, aging, as well as chronic conditions, such as obesity and Crohn’s disease, are characterized by a dysbiosis of the microbiota, distinguished by an increase in proteobacteria, such as *E. coli* [56], which, as mentioned in the introduction, may cause folate-dependent toxicity [13,14]. Interestingly, drugs, such as sulfasalazine, which target bacterial folate synthesis, have proven efficacy treating Crohn’s diseases and ulcerative colitis [57]. Therefore, according to the data obtained here, guarana could potentially represent an interesting diet supplement for these pathologies, owing to its ability to reduce bacterial folate production. Moreover, despite the fact that it remains to be determined whether inhibiting bacterial folate synthesis would slow aging in humans, the combination of guarana’s antioxidant capacity and its inhibition of bacterial folate synthesis seem to make it an interesting anti-aging nutraceutical. 

It has to be additionally taken into account that micronutrients, such as folate, can be obtained from the bacterial microbiota but also directly from the diet [16]. Specifically, vitamin B9 or folate is one of the most commonly supplemented vitamins, as it has been proven to prevent or ameliorate the symptoms associated with its deficiency, such as congenial birth defects [58]. However, in the dosage paradigm, the bacterial folate production and its assimilation into host tissues should be taken into account [16]. This non-predicted source of folate could result in adverse effects, such as zinc deficiency or even an increased risk of colorectal cancer [59,60]. Therefore, given the potential risks of both folate deficiency and excess, it would be interesting to be able to predict the impact of other diet components, such as guarana, in its physiological levels. 

With this study, we managed to further characterize the properties of guarana, confirming its antioxidant and anti-aging capacity, and providing new information on its impact on *E. coli* metabolism. According to the data obtained, the weaker improvement in the lifespan observed in the presence of dead bacteria could be an outcome of the direct beneficial effects of the guarana extract on *C. elegans*. Based on the existing literature and our own results, the antioxidant properties of guarana have a significant role in them. With the current data, we cannot give categorical assurance that the effect observed in the animals fed with dead *E. coli* is due exclusively to the antioxidant effect of the extract or corresponds to the net benefit of its antioxidant effect in the model. In order to clarify this, as part of future work, we will utilize *daf-16* and *skn-1*-deficient mutants to assess the impact of guarana on *C. elegans* lifespan, fed with metabolically inactive *E. coli.* In addition, we plan to determine whether it is one of the major components of guarana responsible for the inhibition of bacterial folate production or if the obtained results are the consequence of the synergistic activity of the complex composition of this plant extract.

Guarana has been extensively studied, and many of its granted properties count with a scientific background; however, there is still hardly any literature on its impact on the gut microbiota [61]. As far as we know, this is the first study showing how the effects of guarana on bacterial metabolism have a direct impact on *C. elegans* survival. The identified capacity of guarana inhibiting bacterial folate production could make it an interesting co-adjuvant in the treatment of intestinal pathologies, characterized by microbiota dysbiosis and increased microbial-derived folate levels. Moreover, given that folate can be directly obtained from diet, bacterial folate synthesis could be targeted without compromising the folate status of the host [24].

Finally, we proved the utility of *C. elegans* as a model to study the potential impact of nutraceuticals on the host health, both directly and indirectly, through their potential impact on bacterial metabolism. 

## Figures and Tables

**Figure 1 nutrients-14-01986-f001:**
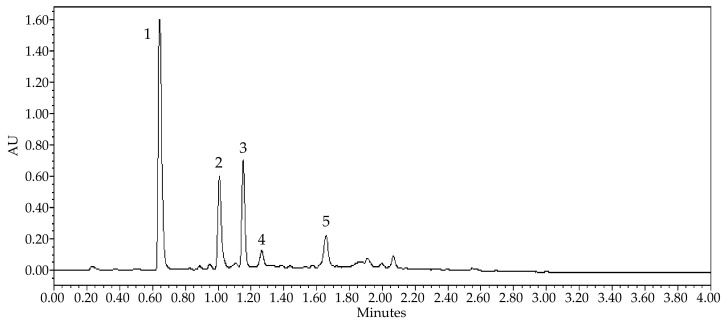
UV Max plot chromatogram of guarana seed extract at 0.0–4.0 min. Key: 1, caffeine; 2, catechin; 3, epicatechin; 4, A-type procyanidin dimer; 5, A-type procyanidin dimer. AU = absorbance units.

**Figure 2 nutrients-14-01986-f002:**
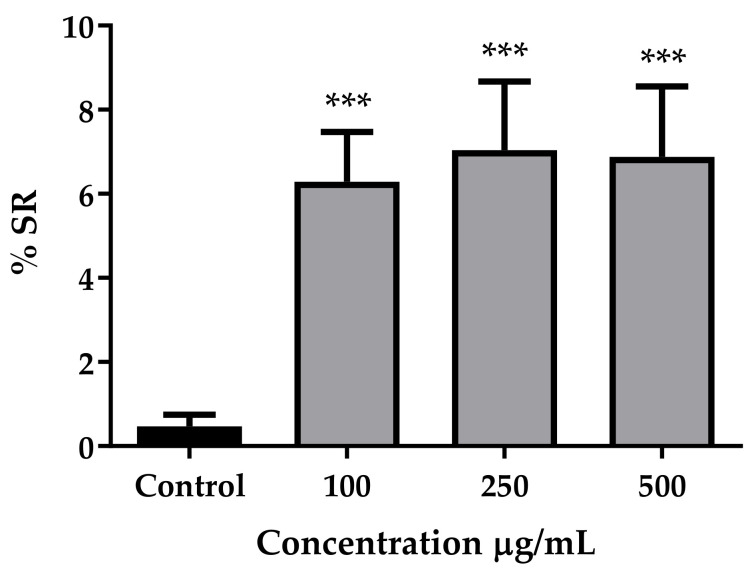
Survival Rate (SR) (%) of *C. elegans* after juglone-induced oxidative stress. The different populations were treated with different concentrations of guarana 100 (*n* = 431), 250 (*n* = 405) and 500 (*n* = 321) µg/mL prior to juglone challenge. Non-treated populations were used as control (*n* = 438). Data are presented as the mean and SEM of survival percentage (*** *p* < 0.0001; Student’s *t*-test).

**Figure 3 nutrients-14-01986-f003:**
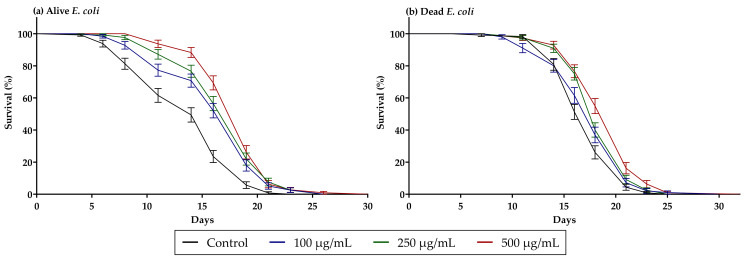
(**a**) Percentage of survival of SS104 *glp*-4 animals (25 °C) fed with live *E. coli* OP50 and treated with different concentrations of guarana, 100 (*n* = 120), 250 (*n* = 119) and 500 (*n* = 111) µg/mL compared with the control (*n* = 125). (**b**) Percentage of survival of SS104 *glp-*4 animals (25 °C) fed with dead *E. coli* OP50 and treated with different concentration of guarana, 100 (*n* = 98), 250 (*n* = 120) and 500 (*n* = 111) µg/mL compared with the control (*n* = 115). Results are expressed as mean ± SD.

**Figure 4 nutrients-14-01986-f004:**
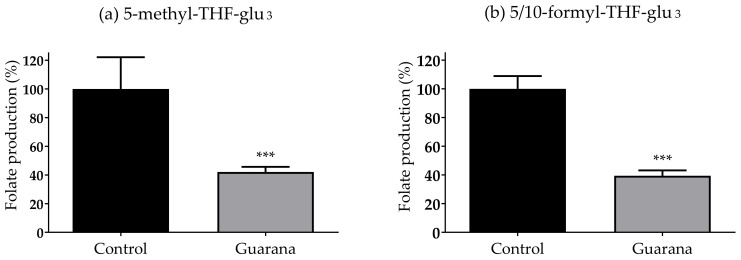
Guarana (500 µg/mL) effects on the *E. coli* production of two folate forms, (**a**) 5-methyl THF-glu_3_ (5-CH_3_-H_4_PteGlu_3_) and (**b**) 5/10-formyl THF-glu_3_ (10-CHO-H_4_PteGlu_3_). Data are presented as the mean and SD of the percentage of folate production of five replicates (*** *p* < 0.0001).

**Table 1 nutrients-14-01986-t001:** Compounds tentatively identified in the guarana seeds extract by UPLC-PDA-MS analyses.

Peak No.	Retention Time (min)	[M − H]^−^	[M + H]^+^	Product Ions		Identified Compounds
				[M − H]^−^	[M + H]^+^	
1	0.64		195.18			Caffeine
2	1.00	289.02		245.00; 203.07; 587.16; 449.06		Catechin
3	1.15	289.02		203.04; 245.00; 220.95; 587.06; 449.01		Epicatechin
4	1.26	575.06		289.03; 284.89; 449.18; 539.11; 557.03; 575.88		A-type procyanidin dimer
5	1.65	574.96		289.00; 575.68; 285.08; 449.00; 423.06; 407.38		A-type procyanidin dimer

**Table 2 nutrients-14-01986-t002:** In vitro antioxidant activity of guarana and epicatechin compared to a positive control, ascorbic acid.

Sample	EC_50_ (µg/mL)
Guarana extract	4.696 to 8.747
Epicatechin	9.320 to 13.780
Ascorbic acid	1.719 to 2.021

**Table 3 nutrients-14-01986-t003:** Average lifespan of SS104 *glp*-4 animals fed with live or dead *E. coli* OP50. Results expressed as days ± SD.

	Average Lifespan (Days)	
Sample	Alive *E. coli*	Dead *E. coli*
Control	13.90 ± 4.40	17.44 ± 2.90
Guarana 100 µg/mL	16.49 ± 4.04	17.84 ± 3.70
Guarana 250 µg/mL	17.47 ± 3.37	18.68 ± 2.86
Guarana 500 µg/mL	18.42 ± 3.19	19.47 ± 0.31

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
