# Peer review of "Alterations in Bacterial Metabolism Contribute to the Lifespan Extension Exerted by Guarana in Caenorhabditis elegans"

_nutrients, 2022, doi:10.3390/nu14091986_

Round 1

Reviewer 1 Report

General objections

A paper entitled: “Alterations in bacterial metabolism contribute to the lifespan extension exerted by guarana in Caenorhabditis elegans” investigates the effect of guarana extract on the lifespan of C. elegans. The authors suggest in their paper that guarana extract can prolong the life of worms not only because of its antioxidant capacity, but also by reducing folate production in E. coli.

In my opinion, the manuscript is incomplete, unclear and does not give any clear conclusion. Specifically, there are insufficient methods that confirm any effect.

  1. First, keywords are inappropriate and the analysis of the main components of the extract is incomplete.
  2. Second, the antioxidant activity of guarana extract was shown only by the DPPH method. For proper assessment of antioxidant capacity, several methods are desirable as antioxidant capacity is based on different principles including reducing power, antioxidant capacity, chelating ability of metals, etc ..
  3. Third, in vivo only GSH was measured which is insufficient for antioxidant efficiency and any conclusion .
  4. Furthermore, there is no link about the importance of folate within our body nor the importance of probiotic strains as a source of folate components. Tetrahydrofolate (THF) or tetrahydrofolic acid is a derivative of Vitamin B9 (folic acid or pteroyl-L-glutamic acid), a water-soluble vitamin that serves as a coenzyme for metabolic reactions involving amino acids and nucleic acids. Folates are required for DNA synthesis and epigenetic regulation. In addition to dietary nutrients, the gut microbiota has been recognized as a source of B complex vitamins, including folate. Folate deficiency has been linked with an increased risk of neural tube defects, cardiovascular disease, cancer and cognitive dysfunction. Exactly, folate is an essential micronutrient that is vital for normal cellular function: adequate folate intake is a critical factor in preventing some neural tube defects (NTD), has been implicated in some forms of anaemia and numerous other adverse health conditions such as cardiovascular disease and cancer. Folate may play a role in the prevention of colorectal cancer while other evidence supports a positive association between increased risk of breast cancer and high folate intake.
  5. What doses are critical for humans? What are the consequences of folate deficiency in humans as guarana alters the metabolism of intestinal bacteria?
  6. The discussion is not focused on the data obtained. Specifically, it describes more the data of other authors without a clear link to the data obtained. The conclusion does not correspond to the obtained results.

Author Response

Please note that the lines indicated in the answers to better follow the additions/modifications are given according to the pdf document. For better reading of the manuscript the track changes are only shown on the word version.

Reviewer 2 Report

This paper reports that guarana extracts extend the lifespan of model organism nematode C. elegans, and the mechanism of elongation may be associated with not only the antioxidant constituents of guarana itself but also alteration of folate metabolism in E. coli. The authors proved the mechanism of longevity mainly using assays for antioxidant activity and LC-MS analysis. Themselves involve partly the original and novel discovery in the efficacy of guarana extracts against C. elegans longevity. However, there are some lacking data and explanation, and unsuitable description containing some incomplete references in their manuscript. The details are mentioned in the following.  

Major comments:

In Results, Figure 4 shows the alteration of bacterial folate metabolism in bacterial plates added guarana extracts. It is revealed that guarana extracts alter the bacterial folate metabolism. However, the authors did not explain about the effects of the alteration of bacterial folate metabolism on the lifespan extension in C. elegans. The authors should explain more concretely how two folate forms (5-methyl-THF-glu3 and 5/10-formyl-THF-glu3) act to the longevity of nematode C. elegans. For instance, like as metformin that inhibits bacterial folate metabolism, whether a component in guarana extracts inhibits or activates the enzymes related to bacterial folate metabolism? Furthermore, how the decreasing in two folate forms (5-methyl-THF-glu3 and 5/10-formyl-THF-glu3) in E. coli compared with control (without guarana extracts) acts to the longevity in C. elegans? Oppositely, two folate forms (5-methyl-THF-glu3 and 5/10-formyl-THF-glu3) have a bad influence on longevity in C. elegans? The authors should explain about them containing the function of folate in C. elegans in Discussion of the manuscript.

As the authors’ argument, if the antioxidant activity in guarana extracts is effective on C. elegans longevity, for example, the authors should confirm whether the extracts (with dead E. coli) extend the lifespan also in daf-16 or skn-1 deficient mutants.

Minor comments:

It’s a pity that there are unsuitable typing and errors such as ‘Error! Reference source not found’ here and there in this manuscript. These descriptions should be revised suitably.

In Results, Table 2 and Figure 2 show the in vitro antioxidant activity of guarana extracts. Because the authors have already measured a variety of antioxidant components in guarana extracts, not only ascorbic acid but also epicatechin etc. should be used for DPPH method as control groups? Perhaps, antioxidant activity of epicatechin etc. is very potent compared with it of ascorbic acid?

Likewise, as shown in Table 2, result of lifespan assays using dead E. coli indicates the net effect of antioxidant in guarana extract?  

In Materials and Methods, the authors describe to utilize the C. elegans two strains N2 and glp-4(bn2) mutant. At least, the authors should explain the reason of using the glp-4(bn2) mutant not wild-type N2 for lifespan assay and about glp-4 gene.

Author Response

(The authors gave the same response as above.)

Reviewer 3 Report

Reigada et al. reported that ethanoic extract of guarana seeds was able to contribute with lifespan extension in C. elegans. The authors show that guarana extract had antioxidant capacity, but it was not correlated with GSH. Moreover, the extract impacted the folate production of E. coli and alteration on E. coli metabolism after extract exposure could contribute to worm life extension.  Although the search using nutraceutical to evaluate its safety and mechanism of action that impacted in microbiota/metabolism are very important, I have some concerns that prevent me to endorse its acceptance at the present stage.

1)Why authors used an ethanolic extracted? Could ethanol change the E.coli metabolism?

2) If caffeine is the major component of the extract, it would be interesting one group using this compound to evaluate the effects of caffeine in E. coli metabolism and observe the effects in lifespan extension.

3) The authors suggest that antioxidant activity of guarana is not involved in GSH system. The analyses of other antioxidant enzymes activities such as SOD, CAT and Gpx should be helpful.

4) It’s hard to follow manuscript, mainly in the results section because there are many “Error! Reference source not found”. This also can be found in discussion section. The manuscript must be carefully revised and reorganized.  

5) The discussion about the influence of microbiota and guarana should be improved.

Author Response

(The authors gave the same response as above.)

Round 2

Reviewer 1 Report

The article has been significantly improved and the authors have respected the suggestions given by the reviewers and have given a lot of effort to give the article quality. Furthermore, the authors also added clarifications on the role of the microbiota and increased folate levels in the body, especially in the elderly population and patients with inflammatory bowel diseases (IBD). Discussion is significantly improved, better related to previous knowledge, but also an emphasis on facts that are still unclear and need to be investigated. I think that some small corrections should be made within the text:

  1. Eliminate the initials of the author within the text (eg Peixoto, H. et al., Reigada I. et al.
  2. Latin names of bacteria eg E. coli write italic
  3. Separate units from numbers
  4. Standardize the way of writing p values (statistical marks should be placed before p and not after p values (instead of p *** <0.0001 should be *** p <0.0001) . Furthermore, for the part of the abbreviation used, the full name should be added and “in vitro” and “in vivo” as well as p value should be written in italics.

Reviewer 3 Report

The authors  answered all comments.

Author Response

We thank the reviewer for the positive feedback.